# A Review of Rehabilitation Benefits of Exercise Training Combined with Nutrition Supplement for Improving Protein Synthesis and Skeletal Muscle Strength in Patients with Cerebral Stroke

**DOI:** 10.3390/nu14234995

**Published:** 2022-11-24

**Authors:** Shiqi Liu, Hengxu Liu, Li Yang, Kun Wang, Nuo Chen, Tingran Zhang, Jiong Luo

**Affiliations:** Research Centre for Exercise Detoxification, College of Physical Education, Southwest University, Chongqing 400715, China

**Keywords:** exercise rehabilitation, protein supplement, nutrition supplement opportunity, oxidative stress, antioxidant, probiotics

## Abstract

Cerebral vascular accident (CVA) is one of the main causes of chronic disability, and it affects the function of daily life, so it is increasingly important to actively rehabilitate patients’ physical functions. The research confirmed that the nutrition supplement strategy is helpful to improve the effect of sports rehabilitation adaptation and sports performance. The patients with chronic strokes (whose strokes occur for more than 6 months) have special nutritional needs while actively carrying out rehabilitation exercises, but there are still few studies to discuss at present. Therefore, this paper will take exercise rehabilitation to promote muscle strength and improve muscle protein synthesis as the main axis and, through integrating existing scientific evidence, discuss the special needs of chronic stroke patients in rehabilitation exercise intervention and nutrition supplement one by one. At the same time, we further evaluated the physiological mechanism of nutrition intervention to promote training adaptation and compared the effects of various nutrition supplement strategies on stroke rehabilitation. Literature review pointed out that immediately supplementing protein nutrition (such as whey protein or soybean protein) after resistance exercise or endurance exercise can promote the efficiency of muscle protein synthesis and produce additive benefits, thereby improving the quality of muscle tissue. Recent animal research results show that probiotics can prevent the risk factors of neural function degradation and promote the benefits of sports rehabilitation. At the same time, natural polyphenols (such as catechin or resveratrol) or vitamins can also reduce the oxidative stress injury caused by animal stroke and promote the proliferation of neural tissue. In view of the fact that animal research results still make up the majority of issues related to the role of nutrition supplements in promoting nerve repair and protection, and the true benefits still need to be confirmed by subsequent human studies. This paper suggests that the future research direction should be the supplement of natural antioxidants, probiotics, compound nutritional supplements, and integrated human clinical research.

## 1. Introduction 

Cerebral stroke is also called stroke and cerebrovascular accident (CVA). It is a group of diseases that cause brain tissue damage, due to sudden rupture of cerebral blood vessels or blockage of blood vessels, including ischemic and hemorrhagic stroke. Most patients with cerebral apoplexy are over 40 years old, and more are men than women, which may cause death in severe cases [1]. According to the results of China’s population census in 2021 [2], compared with stroke patients in 2010, the 10-year growth rate of CVA patients in China is 5.38%, accounting for 22.33% of the total national deaths and ranking third among all causes of death. At the same time, the census also shows that, compared with 2010, the proportion of people over 60 years old in China has increased by 5.44%, accounting for 18.7% of the total population, including 264 million elderly people over 60 years old, which shows that the aging degree in China is intensifying. The epidemiological survey of CVA in Asia shows that the proportion of stroke patients aged over 55 years in Asia accounts for 79.63% of the total stroke patients, and the proportion of patients aged over 65 years is 52.41% [3,4,5]. It can be seen that aging is an important factor to consider when formulating rehabilitation training plans for patients with chronic diseases (especially CVA patients) in the future.

CVA is characterized by high incidence rate, mortality and disability [6,7]. Due to the lack of effective treatment, prevention is currently considered the best measure. Clinical and follow-up studies found that malnutrition and obesity may be the current problems faced by stroke patients. Foley's review article [8] mentioned that the incidence of malnutrition and dysphagia in CVA patients reached 8.2–49.0% and 24.3–52.6% respectively. Schalk et al. pointed out that [9] there is a significant correlation between low plasma albumin concentration and insufficient grip strength in stroke patients due to nutritional deficiencies. Relevant studies have confirmed [10,11] that high protein, especially whey protein or branched chain amino acid supplements, can significantly stimulate the pro-tein synthesis of skeletal muscle and improve the quality of muscle tissue. On the other hand, in the exercise rehabilitation plan for CVA patients, therapeutic exercise is often used as one of the means of rehabilitation to improve patients' physical functions Through planned physical movements and appropriate activities, patients can improve their muscle strength, cardiopulmonary endurance, walking ability and daily living ability, thereby achieving the goal of preventing injury, improving function and improving their overall health. A randomized controlled trial by Yoshimura et al. [12] found that 39 elderly patients hospitalized in the rehabilitation center due to the decline of skeletal muscle quality spent 2–6 months from admission to discharge. Resistance training combined with nutrition supplementation can improve the circumference of the lower leg, upper arm and activities of daily living. Rbadi et al. [13] conducted a prospective, randomized, double-blind controlled study on stroke rehabilitation patients, and they found that in 102 hospitalized patients with acute stroke undergoing rehabilitation treatment, daily intake of nutritional supplements would help promote the recovery of motor function, improve the performance of walking for 2 min and 6 min, and significantly increase the scores of the FIM score and the FIM motor subscale score. Some scholars combine rehabilitation exercise with nutrition supplement, and find that different nutrition supplement strategies are helpful to improve exercise rehabilitation adaptation effects and performance. A supplement of carbohydrate combined with protein or amino acid can significantly stimulate protein synthesis and muscle growth of skeletal muscle [14,15,16,17]

This seems to imply that if the rehabilitation exercise plan of CVA patients can be matched with other alternative therapies (such as special nutrition supplement strategies), it may be more helpful for the rehabilitation of CVA patients.

However, at present, there are few domestic and foreign research literatures on the exercise rehabilitation training intervention nutrition supplement for CVA patients. This paper will review the existing empirical studies systematically, and review the special needs of CVA patients for exercise rehabilitation, the combined effects of exercise rehabilitation and nutrition intervention (amino acids, animal/plant proteins, antioxidants, probiotics, etc.) and the possible physiological mechanisms from a holistic perspective, Thus, it can provide important reference for clinical workers or community caregivers in assisting CVA patients in sports rehabilitation, as well as propose the possible future direction of this topic.

## 2. Data and Methods

### 2.1. Data Sources

This paper systematically analyzes and reviews the latest literature. The authors used computers to search the domestic and foreign research paper databases (including PubMed, Medline, Web of Science, Cochrane, CNKI, and Weibo) from May 2000 to May 2022. The keywords used (including Chinese and English) mainly include: cerebrovascular accident (CVA), stroke, subacute stroke, brain stroke, exercise training, rehabilitation training, protein supplement, timing of nutritional supplement, oxidative stress, antioxidants, probiotics, and balance.

### 2.2. Literature Shortlist Criteria

(1) Subjects are acute or subacute patients who have been ill (stroke) for at least 6 months, that is, they enter the chronic phase 6 months after the occurrence of stroke. (2) The experimental group received aerobic exercise rehabilitation (including resistance in part) and nutrition supplement strategy. (3) The control group only accepted the traditional nutrition supplement strategy. (4) Sports rehabilitation mainly includes endurance training, balance and coordination training, body mechanism and posture stability, flexibility exercise, joint activity exercise, gait training, action training, relaxation skills, and muscle strength training. (5) The evaluation results include: 6 min walking performance, walking speed, stepping strength, lower limb muscle strength, muscle hypertrophy, and sitting to standing time of functional indicators. The indicators of disability degree include Berger’s Balance Scale, Functional Reach Test, and Action Ability.

### 2.3. Literature Exclusion Criteria

(1) Exclude documents whose language is not English or Chinese. (2) Non-randomized controlled trials are excluded. (3) Exclusion of non-exercise rehabilitation combined with nutrition supplement strategy. (4) Patients with CVA less than 6 months were excluded.

### 2.4. Evaluation of Data Intake Quality

(1) The literature was read in three stages. In the first stage, a researcher searches the database, browses the topics and abstracts, and preliminarily screens the documents found. In the second stage, another researcher sorted out the literature and eliminated the duplicate literature. In the third stage, the two researchers jointly read the full text to determine whether the literature meets the inclusion criteria. If there is any literature that has not reached a consensus, it will be decided after discussion.

(2) Literature quality and empirical grade. PEDro Scale is used to check each document to evaluate its research quality and score it. The higher the score, the better the research quality of this document. Each article was scored independently by two researchers. If there were different scoring items, consensus was reached after discussion. Due to of the characteristics of the included papers, the therapists are required to provide treatment intervention during the research process, and the highest total score may be 9 points for the items that cannot be scored by the single blind therapist. Therefore, it is determined that those whose PEDROScale score is greater than or equal to 5 points are high-quality papers, and those whose PEDROScale score is less than or equal to 4 points are low-quality papers. The score of the papers shortlisted in this study is ≥4.

The relevant literature was searched according to the keyword search strategy, 185 relevant pieces of literature were found in total, 75 duplicate pieces of literature were excluded, and 42 pieces of literature inconsistent with the selection criteria were excluded from the remaining 110 pieces of literature after browsing the title and abstract. Among them, the subjects of 9 articles did not meet the inclusion criteria, the evaluation results of 15 articles did not include the requirements of the test items to be used in this study, 12 articles were designed as non-randomized controlled trials, and 6 articles were not in English or Chinese. Finally, 68 articles on randomized control tests were included for systematic review. The process of literature search and inclusion is shown in Figure 1.

## 3. Results

### 3.1. Possible Mechanism of Exercise Training Combined with Nutrition Supplement to Promote Muscle Growth (or Recovery) in Patients with CVA

Shown in Table 1.

Cerebral stroke is caused by the damage of upper neurons. After treatment in the acute phase, survivors may leave sequelae. For example, sensory and motor dysfunction, such as abnormal muscle tone, impaired gait posture control and coordination, masticatory and swallowing dysfunction, cognitive dysfunction, such as aphasia, language disorder, and communication disorder, emotional abnormalities such as depression, and decline of cardiopulmonary endurance, must receive long-term and continuous rehabilitation treatment to reduce disability and restore normal life function [34,35,36]. Once limb hemiplegia occurs in CVA patients, the physical activity of the patients will be significantly reduced, and the decline in physical activity is closely related to muscle atrophy. Skeletal muscle is an important organ to maintain constant blood glucose in the human body. Under the stimulation of insulin, about 85% of glucose is absorbed and stored by skeletal muscle. It is known that regular physical activity can increase muscle quality, improve skeletal muscle microvessel density and blood perfusion, increase muscle GLUT4 content, and further improve skeletal muscle insulin sensitivity and muscle glucose absorption capacity [37]. Atrophy of paralyzed muscles will lead to atrophy of blood vessels in muscle tissue, thereby reducing the blood flow of paralyzed parts (about 35% lower than that of ordinary people). The glucose absorption capacity of muscle tissue is directly regulated by the carbohydrate absorption capacity of cells and the blood perfusion of tissues, so vascular atrophy will further limit the carbohydrate absorption capacity of paralyzed muscles [38]. After a stroke, the muscle fiber shrinks, the muscle starting speed decreases, the second kind of muscle fiber atrophies and is easy to fatigue, the number of action units decreases, and other action units are recruited, which makes the patient have muscle weakness and affects the movement control, walking speed, and endurance. 

#### 3.1.1. mTOR Protein and Its Mechanism of Action

MTOR Protein (mammalian target of rapamycin, mTOR) is a serine/threonine kinase. The activity of mTOR depends on its phosphorylation degree. It is known that mTOR has three phosphorylation sites, namely Ser2481, Ser2448, and Thr2446. When the above positions are phosphorylated, mTOR will immediately start to activate its downstream target factors (downstream target factors: 4E-BP1 (eucaryotic initiation factor 4E binding protein 1), eIF4G (eucaryotic initiation factor 4E), and p70s6k (p70 ribosomal S6 kinase)) [39,40], thereby promoting protein synthesis. It was found that mTOR protein was activated by protein kinase B (PKB/Akt), a downstream signaling factor of insulin [41]. In addition, the activation of mTOR is not only directly through the PKB/Akt signaling pathway but also affected by TSC1 and TSC2 (ambient fibrosis complex 1,2) proteins. When TSC1 and TSC2 combine to form a complex, it will significantly inhibit the activation of mTOR protein. However, when PKB protein is activated, TSC2 will be phosphorylated, thus separating the TSC1/TSC2 complex and finally reversing the inhibition effect on mTOR performance (see Figure 2).

#### 3.1.2. Exercise Training and Activation of mTOR Protein

Rehabilitation exercise training, including resistance training, endurance training, functional training, and stretching training, can significantly increase the synthesis of skeletal muscle protein in the recovery phase after training [22,32]. Relevant studies have shown that skeletal muscle protein synthesis is mainly achieved by activating the PI3K/PKB/mTOR (physiologically linositide 3-kinase/PKB/mTOR) pathway [27,28]. After the old rats were given weight bearing exercise or used electric stimulation to cause muscle contraction, the activation level of PKB and the phosphorylation level of mTOR protein could be significantly improved, and the phosphorylation of GSK3 (glycogen synthase kinase 3) could also be promoted [24,33]. It is found that GSK3 is another protein that can regulate muscle hypertrophy. GSK3 protein will be phosphorylated by PKB protein, which will inhibit its own activity, thus activating eIF2B (eucaryotic translation initiation factor 2B) and, finally, increasing muscle protein synthesis. After rehabilitation exercise training, the activity of eIF2B will be significantly improved, so the effect of improving activity is consistent with that of increasing muscle protein synthesis [29,42]. In conclusion, at present, more and more evidence points out that exercise training can increase the protein synthesis of muscle cells and improve muscle quality, mainly by activating the mammalian rapamycin target protein (mTOR) information system in muscle cells so that the ribosome can improve the protein translation efficiency and further increase the skeletal muscle protein synthesis ability [31]. Moreover, the phenomenon of muscle protein synthesis caused by a single exercise can last for several hours [43]. Although the rate of muscle protein synthesis will increase within 1–2 h after exercise, long-term exercise will also cause the oxidation of amino acids in the body, thereby antagonizing the benefits of exercise stimulating protein synthesis.

#### 3.1.3. The Physiological Mechanism of Nutritional Supplement, Amino Acid Circulation and Muscle Growth

The possible benefit mechanism of combining the particularity of exercise training for CVA patients with the existing nutritional supplement methods, to improve or promote the functional exercise ability of stroke patients, can be summarized in the following aspects.

(1) The intake of a protein-rich diet helps to improve the bioavailability of amino acids in the circulatory system and can directly stimulate the rate of muscle protein synthesis [44,45,46]. It is known that dietary protein intake can improve training adaptation benefits and promote muscle strength [23,47], so it has been considered as a common strategy to maintain muscle quality of the elderly [48]. In normal protein metabolism, adequate nutrient intake can increase the bioavailability of large amounts of nutrients and compensate for the loss of essential amino acids caused by protein oxidation during normal metabolism.

(2) In addition to providing amino acids as a source of protein synthesis in muscle cells, certain specific amino acids (such as branched chain amino acids, especially leucine) can also be used as intracellular stimuli to promote protein synthesis in muscle cells by activating the mTOR information system [49,50,51,52]. The supplement of carbohydrate combined with protein or amino acid immediately after exercise can significantly stimulate the protein synthesis and muscle growth of skeletal muscle, which has the benefit of improving the training adaptation effect and sports performance [18,19].

(3) Supplying protein and carbohydrate at the same time can rapidly improve the pancreas β Cells additively release insulin [53,54]. As insulin is an important synthetic hormone, it will contribute to the energy and material synthesis of various cells. Research shows that providing protein and carbohydrates immediately after exercise can significantly stimulate muscle protein synthesis and muscle growth [35]. The research results also emphasize that the theory of nutrient intake time plays a key role in training adaptation benefits caused by exercise [20,55]. In addition, insulin can also promote the absorption capacity of muscle cells to various nutrients and can also slow down the proteolysis of skeletal muscle induced by exercise [26,55].

In conclusion, the sensitivity of the protein synthesis system of muscle cells to nutrients has also been significantly improved after exercise, and these effects are mainly regulated by changing the nutrients in the blood (such as amino acids), the content of synthetic endocrine hormones, and the protein synthesis system of cells. Therefore, relevant nutrients can be supplemented at an appropriate time, and it seems that it can improve the rehabilitation effect of CVA rehabilitation exercise by providing high-quality protein bound carbohydrates.

### 3.2. Analysis of the Positive Effect of Exercise Training Combined with Nutritional Supplement on the Rehabilitation of CVA Patients

Shown in Table 2.

#### 3.2.1. Rehabilitation Effect of “Carbohydrate and Protein Nutrition” Supplement on CVA Patients

(1) Muscle fiber hypertrophy and muscle mechanics. When the long-term exercise rehabilitation is completed, it will be accompanied by muscle damage, amino acid oxidation in the body, protein imbalance, etc. Supplementing carbohydrate drinks containing protein after exercise can change the nitrogen balance from negative to positive, indicating that it is helpful to promote muscle protein synthesis [57]. In addition, during the recovery period after endurance exercise (75% VO_2max_), it was proved that the supplementation of protein sugar drink (carbohydrate: proteiN = 4:1) to the subjects could improve the net synthesis of muscle protein and alleviate the muscle injury caused by endurance exercise [21]. Dreyer et al. [61] found that the protein synthesis rate increased by three times compared with that at the end of exercise, when protein plus carbohydrate beverage was supplemented one hour after moderate intensity exercise, but if it was supplemented three hours after exercise, the protein synthesis rate only increased by 12%. It has also been found that intravenous infusion of amino acid (supplementation of serotonin: 0.15 g/kg/h or supplementation of hydrolyzed whey protein/leucine/carbohydrate) after resistance exercise can accelerate skeletal muscle protein synthesis [30,62]. Similarly, some scholars have conducted resistance training for 14 weeks and immediately added protein or carbohydrate before and after each training (note: supplement in the morning on days without training). At the end of the whole intervention cycle, they measured the cross-sectional volume of muscle fibers of the lateral thigh muscle, deep squat jump, squat jump, isokinetic muscle strength, and rapid maximum torque of the subjects. The results showed that type I hypertrophy of type II muscle fibers was obvious, and the height of squatting jump was significantly increased, while there was no significant change in the carbohydrate group [63].

(2) In terms of muscle strength and functional activity. In a rehabilitation training experiment for the disabled elderly [58], the subjects were divided into resistance training group, nutrition supplement group, resistance and nutrition combination group, and control group. The resistance training group received progressive resistance training of lower limbs (Three times a week, lasting for 10 weeks, with 45 min of training each time, and one day of rest in between; all training sessions are individually guided by a certified therapeutic entertainment specialist), the nutrition supplement group only received protein supplement, and the resistance and nutrition combination group received the same progressive resistance training of lower limbs and protein supplement. The control group only received entertainment therapy (playing games, concert, and group discussion). The results showed that the lower limb muscle strength, walking speed, and stepping strength of the resistance training group increased by 113%, 11.8%, and 28.4% respectively, which were significantly higher than those of the nutrition supplement group and the control group. The resistance and nutrition combination group is significantly superior to the simple resistance training group in balance ability and muscle strength improvement. In terms of nutrition supplement time selection, although a large number of studies have confirmed that protein intake immediately after resistance training is more conducive to skeletal muscle hypertrophy in elderly CVA male patients, it has not been confirmed whether it can effectively restore the follow-up exercise ability to the previous level. For example, Rustad et al. [64] discussed the effect of exercise training from treadmill to exhaustion and randomly gave the subjects protein plus carbohydrate or only carbohydrate as supplements within two hours (each subject performed three times of exhaustive exercise and received three different nutritional supplements after exercise: 1.2 g carbohydrate/kg/h, 0.8 g carbohydrate + 0.4 g protein/kg/h, and placebo (no calories). The results showed that carbohydrate and protein supplementation could increase blood glucose, insulin, and BCAA, while carbohydrate supplementation only increased blood glucose and insulin. After 18 h, the subjects conducted another “time to exhaustion” (TTE) experiment. Compared with those who only received carbohydrate supplements, the TTE of those who received carbohydrate and protein supplements increased significantly.

To sum up, when CVA patients receive clinical exercise rehabilitation treatment, they can improve their daily functional activity ability through endurance training, resistance training, balance training, and other measures. If these exercise rehabilitation methods are combined with nutrition supplement strategies, it will be more helpful to improve the functional exercise performance of CVA patients. However, after resistance or endurance sports training, the timing of supplement of nutrients should be concerned. If protein or amino acid is supplemented immediately, it can significantly promote the efficiency of muscle protein synthesis and further improve muscle quality. However, only a carbohydrate supplement after training has no significant effect on the training effect.

#### 3.2.2. Rehabilitation Effect of “Amino Acid and Protein Nutrition” Supplement on CVA Patients

Rabadi et al. [13] found that, during the clinical rehabilitation period of malnourished patients with acute CVA, three times of daily nutrition supplement (5 g protein/time) will help to promote the recovery of action function. Esmarck et al. used progressive resistance exercise (3 times/week, 12 weeks in total) with nutritional supplement (10 g of milk/bean protein) to compare the efficacy of immediate supplement after exercise with that of 2 h supplement after exercise. The results showed that the cross-sectional area of quadriceps femoris, in the immediate supplement group after exercise, increased by 6.77%, and the average muscle fiber range increased by 24.01% [60,65]. In terms of the difference in the effects of supplementing different protein types, the relevant research found that [66], compared with the equal calorie formula containing casein, the enteral nutrition formula containing whey protein can more effectively reduce the systemic inflammatory reaction of elderly patients with acute stroke and increase the antioxidant capacity, which proves that the ability of whey protein in anti-inflammation is significantly better than that of casein. However, soy protein supplementation can accelerate the muscle protein synthesis rate, which is close to whey protein supplementation and is even better than casein supplementation [67,68]. On the other hand, Scherbakov et al. [56] studied that, during hospitalization, patients with acute CVA were given essential amino acid supplements three times a day (4 g essential amino acid/time). After 8 weeks, they did not find that this nutritional supplement strategy could significantly improve the muscle atrophy of CVA patients. Carlsson et al. [59] also found that the elderly did not significantly promote the increase in muscle strength or functional recovery when adding protein.

Based on the above results, the effect of exercise rehabilitation combined with “amino acid plus protein nutrition” for CVA patients is still controversial at present. The reason may be that different researchers use different nutritional supplement doses, time nodes, etc., and it may also be related to the difference in resistance strength. However, most of the empirical results for patients with acute CVA seem to be inclined to exercise rehabilitation combined with “soy protein or whey protein”, which seems to be more helpful to improve their muscle mass, muscle strength, reduce inflammatory reaction, as well as improve antioxidant capacity and rehabilitation exercise adaptation benefits. However, at present, there are few studies on this aspect, and more clinical rehabilitation experiments need to be carried out by future scholars.

#### 3.2.3. Rehabilitation Effect of “Probiotics or Natural Antioxidants” on CVA Patients

At present, relevant theories show that the gastrointestinal tract is connected with the central nervous system through the brain/intestinal axis, thus supporting the development and maintenance of neurons. Therefore, the development of nervous system diseases has been found to be highly related to gut dysbiosis [69]. Obviously, aging is a common reason for the development of degenerative neurological diseases. The study found that [70] probiotics supplementation can increase the concentration of neurotransmitters, inhibit systemic inflammation, reduce the oxidative stress of the nervous system, and so on. Therefore, it can be used as a supplement to prevent aging from degrading neural function. Bel Rhlid et al. found [71] that the combination of probiotic strains and plant-derived polyphenol extracts can be used to treat or prevent degenerative neurological diseases or Alzheimer’s disease. Chen et al. [72] confirmed that supplementing lactobacillus plantarum (TWK10) can improve the exercise tolerance and increase the muscle mass of mice. In terms of antioxidant supplement, the study found that the plasma vitamin C concentration of CVA patients decreased significantly, accompanied by systemic inflammatory reaction, which means that the antioxidant capacity of CVA patients decreased significantly [73]. Excessive inflammatory reaction will cause cell damage and apoptosis, which may have a negative impact on the prognosis of stroke and may further induce the development of depression in the prognosis of CVA patients [74]. Relevant studies have found that natural polyphenol compounds such as catechin or resveratrol can reduce oxidative stress after stroke in animals [75] or cell oxidative damage caused by physiological pressure [25]. The supplement of antioxidant polyphenols can promote the proliferation of neural tissue [76].

Reviewing the recent literature, we suggest that, compared with simply supplementing probiotics or polyphenol antioxidants, combined with rehabilitation exercise specifically for patients with chronic stroke, it may have a better effect on reducing oxidative stress and slowing down neural/motor function degradation. However, the above studies on nerve injury or stroke are mostly the results of animal experiments, so the actual benefits still need to be confirmed by subsequent human studies. The change of intestinal microbiota may be closely related to some neuropathies. Adjusting the distribution of intestinal microbiota by adding probiotics is helpful to prevent or alleviate degenerative neuropathy. Although there is no research evidence to explore whether probiotic supplementation can promote the recovery of neurological function in patients with chronic stroke, recent cumulative research also shows possible benefits. These research results all show that the application of probiotics has potential benefits for promoting the recovery of neural function or sports ability in combination with sports rehabilitation. Therefore, a supplement of antioxidants may be a feasible strategy to help reduce oxidative stress and systemic inflammatory reaction in patients with chronic stroke.

In a word, most of the existing studies on rehabilitation training and nutrition supplement for stroke patients focus on the acute period after stroke (about one month during acute hospitalization). The important part of this stage is to ensure that stroke patients can obtain enough nutrition supplement to maintain their physical health and function so that they can have enough physical strength to start acute stroke rehabilitation treatment at an early stage. At present, many studies have pointed out that, through constructive and closely monitored exercise rehabilitation (3–4 times a week, lasting for 4–13 weeks), the functional indicators of patients with chronic stroke can be effectively improved (promoting 6-min walking performance, walking speed, step strength, lower limb muscle strength, muscle hypertrophy, reducing the time required for sitting to stand) and the degree of disability (improving mobility indicators). More importantly, after the chronic stroke patients leave the hospital and returns to their families and communities, it is the beginning of active rehabilitation exercise training and promoting functional recovery. At this time, more active rehabilitation exercise training intervention should be combined with appropriate nutrition intake to help patients obtain the maximum rehabilitation treatment benefits. In addition, most of the chronic stroke patients are the elderly, according to the relevant research of the elderly, so it is pointed out that the exercise training effect can be maximized only by integrating appropriate nutrition intervention at appropriate training time. However, improper training intensity or mode may lead to the increase in oxidative stress, which leads to poor training results. Based on the above evidence, the benefit of combining carbohydrate/protein supplement during exercise training, to improve the muscle strength and promote muscle hypertrophy of the elderly, is enough to provide reference for clinical workers or community caregivers to assist patients with chronic stroke in planning exercise nutrition strategies.

### 3.3. Analysis of Negative Effect of Exercise Training Combined with Nutrition Supplement on CVA Patients’ Rehabilitation

Walking is one of the most commonly used rehabilitation methods for CVA patients. However, even the simplest walking exercise may bring negative benefits to some CVA patients if the prescription intensity is not appropriate [77]. The lower limb muscles are in continuous stretching/shortening (eccentric centripetal, that is, SSC contraction) contraction during walking, so this rapid muscle contraction mode will lead to muscle damage, but long-term SSC training will significantly improve the performance and muscle hypertrophy of young adult rats [78]. However, when carrying out the same SSC training for aging rats, it will lead to decreased exercise performance and inflammatory reaction of cells. If the duration is longer, it may lead to inadaptable effects such as muscle function loss and muscle mass loss [79]. The latest research shows that [80], the older the subjects are, the more systemic inflammatory reaction and higher oxidative pressure will occur during a long period of moderate intensity muscle contraction speed and intermittent SSC training, which will lead to poor recovery ability of muscle injury and discomfort in muscle strength training. Sanchez Moreno et al. [73] discussed the relationship between the basic antioxidant capacity and the inflammatory response of CVA patients. The results showed that the plasma vitamin C concentration of CVA patients decreased significantly, accompanied by a significant increase in serum inflammatory indicators (C-reactive protein, etc.) and the concentration of oxidative pressure markers, indicating that there was a significant correlation between the decline of systemic antioxidant capacity of CVA patients and stroke related inflammatory response. Although a large number of studies have shown that exercise combined with nutrition supplement is beneficial to promote muscle growth or increase muscle strength, in the face of CVA patients, it may be affected by factors such as muscle degradation or muscle disuse, resulting in passivation reaction [29,80,81] (also known as “anabolic impedance phenomenon”), which makes muscles unable to maintain their muscle protein content by properly stimulating protein synthesis and leads to sarcopenia in the process of degeneration. It is also pointed out that, even though CVA patients receive rehabilitation treatment, there are still about 30–60% of patients whose motor function has not improved significantly, indicating that conventional rehabilitation training may have shortcomings. The reasons can be summarized as follows: (1) during the recovery of CVA prognosis, the regular rehabilitation exercise cannot obtain sufficient motor stimulation; (2) the increase in oxidative stress and inflammation caused by aging may affect the effect of muscle strength training; (3) aging factors cause the phenomenon of anabolic impedance.

In a word, in view of the fact that CVA patients generally suffer from muscle fiber shrinkage, reduced muscle starting speed, type II muscle fiber atrophy, fatigue, and reduced number of action units, they often feel muscle weakness during exercise training, which further affects action control, walking speed and endurance, plus the interaction of aging factors. Therefore, when CVA patients receive rehabilitation exercise training, Muscle tissue will bear more physiological pressure, which may lead to the phenomenon that the training effect will not rise but fall. Based on the possible negative effects of sports rehabilitation, it may be necessary to start with a mild and gradual training mode for special groups such as CVA and adjust the degree of systemic inflammation and oxidative stress, so they can get a better training adaptation environment first and, then, start to adjust the intensity of sports rehabilitation as to avoid poor adaptation of muscle strength training. At the same time, in addition to effective and highly monitored exercise training intervention, it may still be necessary to integrate other non-drug active intervention methods (nutrition supplement strategy) to help reduce oxidative stress and chronic inflammatory reaction of the system, slow down anabolic impedance reaction, etc., which is extremely important for CVA patients to maintain their daily independent living ability by increasing muscle volume and strength.

## 4. Conclusions

(1) Most CVA patients are elderly. In the early stage of rehabilitation training (one month before acute hospitalization), it is a key core task for patients to obtain adequate nutrition supplement. At the same time, appropriate exercise training time combined with appropriate nutrition supplement can inhibit excessive oxidative stress so as to maximize the effect of exercise rehabilitation.

(2) Proper exercise prescription and strict training monitoring can effectively improve the 6 min walking performance, walking speed, step strength, lower limb muscle strength, muscle hypertrophy, reduce the time required for sitting to stand, reduce the degree of disability, and other functional indicators of CVA patients.

(3) Rehabilitation exercise training combined with carbohydrate/protein supplement can significantly improve muscle strength and promote muscle hypertrophy in patients with acute CVA (prognosis: 6 months). However, the positive effect of exercise rehabilitation combined with an “amino acid plus protein nutrition” supplement on CVA rehabilitation is still controversial. Although there is no empirical study indicating that probiotic supplements can help promote the recovery of neural function in CVA patients, antioxidant supplements may help reduce the oxidative stress and systemic inflammatory reaction in CVA patients.

## 5. Problems and Prospects

(1) The injury of an upper motor neuron in CVA patients leads to paralysis, and some skeletal muscle atrophy may lead to functional defect and disability. In addition, after active treatment in the acute and subacute period, a large number of survivors still have an obvious disability, so CVA patients have obvious physical decline, lack the ability to perform daily life functions, and may fall into a vicious circle of disability and decreased mobility. Therefore, the active rehabilitation treatment and sports intervention provided to CVA patients to promote functional recovery and prevent subsequent deterioration of health status will help reduce the possibility of subsequent cerebrovascular diseases

(2) The rehabilitation training of CVA patients should consider many special neuromuscular relationships, so there is a big difference between the training purpose and mechanism of the general population, the elderly, or athletes. CVA patients have many different indicators of disability, such as limb movement, language expression, swallowing function, cognitive function, psychological depression, etc., but considering the muscle function and muscle strength required to maintain limb movement is also the focus of rehabilitation to perform daily movement functions.

(3) In the rehabilitation exercise training plan adopted by CVA patients, at present, it is mainly embodied in endurance training, balance and coordination training, body mechanism and posture stability, flexibility exercise, joint range of motion, gait training, movement training, relaxation skills and muscle strength training. Therapeutic exercise classes help with movement control, upper and lower extremity use, mobility, balance, and aerobic exercise. The content of sports rehabilitation covers a wide range, such as joint activities (passive, assisted active, active), progressive muscle resistance training, endurance training, neuromuscular induction technology, sports function training, endurance sports using sports rehabilitation equipment (bicycle, treadmill, handcart), etc.

(4) At present, many studies have shown that the functional indicators of CVA patients can be effectively improved through constructive and closely monitored exercise rehabilitation prescriptions (3–4 times a week for 4–13 weeks). On the other hand, within 6 months, from the onset of stroke to discharge from hospital, about 41% of stroke patients showed the risk of malnutrition, mainly due to the negative effects of low appetite, dyspepsia, daily activity ability, and dysphagia. In addition, many patients with chronic stroke are also elderly people. Research also shows that there is a clear correlation between low plasma albumin concentration caused by insufficient nutrition intake and insufficient grip strength. These factors may lead to poor rehabilitation training.

(5) In CVA patients, the imbalance of nutrient intake not only limits physical performance, but also tends to develop into a high prevalence of obesity and intramuscular fat infiltration in the affected muscle group in the chronic phase. This shows that CVA patients can cause imbalance in energy consumption and calorie intake, as well as accelerate the development of obesity, due to the significant reduction in activity and muscle loss. These negative changes will further damage the cardiorespiratory fitness and muscle activity. Therefore, insufficient caloric intake, excessive caloric intake, and poor diet quality will increase the follow-up nutritional risk of patients with chronic stroke and may hinder the functional recovery of stroke patients. It can be seen that sports rehabilitation training programs should include appropriate nutrition interventions to improve the functional recovery of patients.

(6) At present, there are few studies on the promotion effect of rehabilitation exercise training combined with proper nutrition supplement on functional indicators for CVA patients, so the relevant conclusions need to be confirmed by more subsequent studies. Although antioxidant and probiotic supplements seem to have a positive effect on promoting neural repair and reducing inflammation, it must be indicated that these studies are mostly based on the results obtained from animal research models. The future application to stroke patients still needs to be verified by follow-up human clinical studies. Therefore, regarding the supplement of antioxidants and probiotics, combined with the clinical benefit verification of rehabilitation exercise training, it is urgent for the academic community to develop corresponding safety standards and clinical evaluation strategies so that clinical staff can effectively assist CVA patients to improve physical fitness and promote functional pointer recovery.

(7) At present, there are many research evidences that nutrition intervention can help to reduce oxidative pressure, promote the proliferation of nerve tissue, improve muscle strength, and promote muscle hypertrophy. Therefore, in the process of rehabilitation exercise, if the nutrition supplement intervention strategy can be properly integrated, it may help to further improve or maintain the physical function and activity of chronic stroke patients, thereby reducing the burden and pressure on caregivers, and in the long run, it can improve the quality of life of caregivers and patients. In the second part of this review, we will integrate the benefits and possible physiological mechanisms of nutritional supplement strategies that may help to reduce oxidative stress and systemic inflammation, as well as slow down the anabolic impedance response in the following sections. These results indicate that aging may weaken the adaptability of skeletal muscle to repeated mechanical load even if there is no obvious degeneration of muscle cells. At present, there is no relevant research on SSC in stroke patients, but a high proportion of stroke patients are elderly. Therefore, in planning exercise rehabilitation for stroke patients, it is also necessary to consider the possible negative effects of training.

## Figures and Tables

**Figure 1 nutrients-14-04995-f001:**
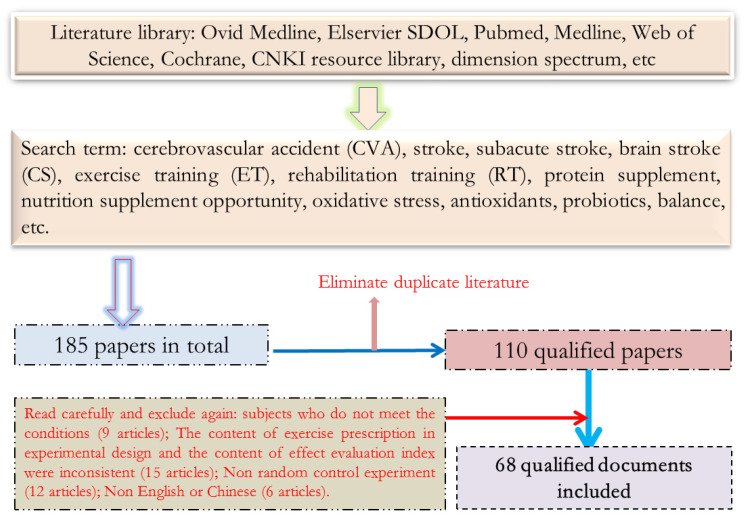
Schematic Diagram of Document Screening.

**Figure 2 nutrients-14-04995-f002:**
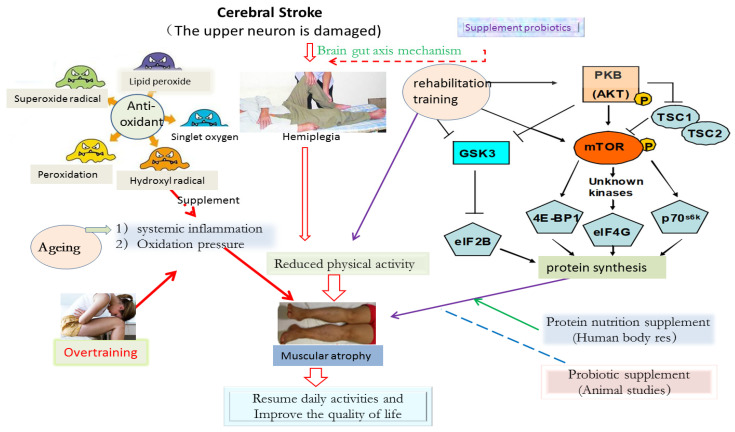
Possible mechanism of exercise training combined with nutrition supplement to promote the rehabilitation of CVA patients.

**Table 1 nutrients-14-04995-t001:** Literature on the rehabilitation effect and influence mechanism of exercise training intervention on patients with chronic stroke.

Author/Year	Test Group (Number)	Sports Training Program	Effectiveness Evaluation	Research Results
Moreland et al., (2003) [18]	*n* = 133 Chronic stroke patients	EG: Resistance training, 10 times/group, 5 groups, 3 times/week, 2 months in total; CG: Traditional treatment.	Lower limb muscle strength, walking for 2 min	lower limb muscle strength ↑ (EG > CG); LLFDI (EG ↔ CG); 2 MW (EG ↔ CG);
Cramp et al., (2006) [19]	*n* = 10 Chronic stroke patients	Self controlled experiment. Resistance: weight, elastic band and body weight; Mode: 5 groups of exercises on the hips and knees of both feet; Intensity: 20% 1 RM first, 50% 1 RM later, 10 times, three groups; Frequency: 2 times/week, 6 months in total.	Lower limb muscle strength; Isometric torque	lower limb muscle strength ↑ Knee extensor muscle strength ↑ Knee flexor muscle strength ↔
Flansbjer et al., (2008) [20]	*n* = 24 Chronic stroke patients	EG: The training instrument is set to 80% of the maximum strength; CG: General daily activities. 90 min/course of treatment, 2 times/week, 10 weeks in total.	Muscle strength and muscle tension; TUG, 6 MW, SIS	↑ Strength in PRE (PRE > CG) ↓ Muscle tone in PRE and CG ↑ Strength in PRE ↑ 3 measures in PRE and ↑ TUG and 6MW in PRE ↑ SIS (EG > CG)
Twist & Eston (2009) [21]	*n* = 15 Chronic stroke patients	EG: pull the elastic band on the upper limb, and make 4 TRTs in different directions respectively; Reach for objects 200 times, 45 min/time, 3 times/week, 4 weeks in total.	Trunk and Arm Motion: AROM, WMFT, FMA	↑ Elbow extension in both groups ↑ Hand path straighten in TRT ↓ Trunk movement and ↑ AROM and FMA in both groups; WMFT:(EG ↔ CG)
Murton & Greenhaff (2010) [22]	*n* = 42 Chronic stroke patients	EG: 70% 1 RM with both feet, repeat 8~10 times as a group, and do three rounds; CG: ROM and soft exercise, 3 times/week, 12 weeks in total.	Lower limb muscle strength; 6 MW, step climbing and station arrival; LLFDI, GDS and SIP	↑ EG lower limb muscle strength; ↑ Two groups of 6 MW and walking speed; ↑ CGLLFDI; ↑ Two groups of GDS; SIP (EG ↔ CG).
Kim et al., (2013) [23]	*n* = 36 Chronic stroke patients	Ankle muscle strength training group VS Lacing group VS CG. First, use 30% of the force to do plantar flexion for 20 times, and then use bare hands to give pressure to do isometric contraction of back flexion. The resistance movement of the ankle joint on the affected side of EG gradually increased by 40–80%. 4 times/week, 6 weeks.	Gait and balance function detection; 6 MW.	The swing amplitude of body pressure center in the three groups increased significantly. Physical function: the experimental group had positive effects on the toe off movement during walking
Lee et al., (2016) [24]	*n* = 36 Chronic stroke patients	Randomized controlled trial. EG: Air lock. Hip, knee and ankle flexion is performed on both lower limbs on the machine. First do 25% 1RM, four times, then do 70% 1 RM, 8–10 times, three groups; 5 times/week, 6 weeks in total.	Gait and balance function detection; 6 MW; BBS; Stand up and walk test	The balance ability of EG and CG increased significantly. EG’s left and right displacement before and after the body's center of gravity, BBS and the results of timed stand walk test were significantly improved
Kuo et al., (2015) [25]	*n* = 40 Chronic stroke patients	EG: Resistance exercise increased muscle strength FT, and repeated functional action control group; CG: Traditional treatment. Frequency and time: 1 h/day, 5 days/week, lasting for 4–6 weeks.	Isometric torque, MotorFMA, FTHUE, and FIM	↑ Isometric torque, motor FMA and FTHUE (EG > CG, EG > CG). Follow up: ↑ Palmar pinch (PRE > CG, FT > CG), AROM (EG ↔ CG).
Yingying et al., (2016) [26]	*n* = 48 Chronic stroke patients	EG: Gradually increase the number of repetitions of six tasks, and CG does not intervene; 30 min/day, 3 days/week, 4 weeks in total.	Lower limb muscle strength; Gait performance, 6MW; Steptest; TUG.	↑ Strength in all muscles in EG; ↑ Gait velocity, 6 MW, and TUG in EG; ↓ Step test in CG + Relationship between muscle strength and function.
Lane et al., (2017) [27]	*n* = 7 Chronic stroke patients	EG: 70% of 1RM was performed on both feet, 8~10 times per group, 3 groups, 2 times per week, a total of 12 weeks	Muscle strength, MAS, BBS; TUG and step registration time; GDS	↑ Muscle strength, MAS, BBS; ↓ Time required for sitting to the station; No difference in GDS
Hasegawa et al., (2018) [28]	*n* = 48 Chronic stroke patients	EG: Gradually increase the number of repetitions of six tasks; CG: no intervention, 30 min/day, 3 days/week, 4 weeks in total.	Lower limb muscle strength; Walking ability; 6 MW, TUG	↑ EG lower limb muscle strength; ↑ EG walking speed, 6 MW, TUG
Kido et al., (2018) [29]	*n* = 92 Chronic stroke patients	EG: ROM, muscle strength strengthening, balance, upper limb endurance and functional training; 90 min/time, 3 times/week, 14 weeks in total; CG: No motion.	Grasping power, FMA, WMF, BBS; 10-Meter and 6-MinuteWalk.	↑ Grip strength, BBS, WMFT in both; ↑ Gait velocity in both. However, EG in each assessment is more improved than CG.
Bauer, et al., (2018) [30]	*n* = 42 Chronic stroke patients	EG: do 70% of 1 RM with both feet, 8~10 times/group, 3 groups/time; CG: ROM and soft exercise, 3 times/week, 6 week in total	Lower limb muscle strength; 6 MW; Stair climb and Chair rise; LLFDI; GDS and SIP.	↑ Strength in most muscles in EG ↑ 6 MW and gait velocity in both ↑ LLFDI in PRT ↑ GDS and no SIP in bot
Williamson et al., (2018) [31]	*n* = 12 Chronic stroke patients	EG: pull the elastic band on the upper limb, and perform 4 different movements TRT respectively; Reach for objects, 150–180 times/30 min, 3 times/week, 4 weeks in total.	Arm motion track, Trunk and ArmMotion, RMA, MAS	↑ Elbow extension in EG ↑ Hand path straighten in TRT ↑ Trunk movement in EG ↑ RMA in low level TRT (No difference for the MAS)
VanDerwerker et al., (2018) [11]	*n* = 24 Chronic stroke patients	EG: the machine is set at 80% of the maximum force; CG: general daily activities, 90 min course of treatment, 2 times/week, 10 weeks in total	Muscle strength, muscle tension, TUG, 6 MW	↑ EG Muscle strength > CG; ↓ Two groups of muscle tension. Follow up: ↑EG Muscle strength; EG: ↑ TUG, 6 MW; ↑ EG: SIS > CG
Smeuninx, et al., (2020) [32]	*n* = 170 Chronic stroke patients	EG: 13 week rehabilitation physical therapy exercise training resistance training, 10 times/group, 4 groups, 3 times/week, 8 Weeks in total; CG: Traditional treatment.	Action capability indicator; Travel speed; Falls	↑ Action capability indicators (EG > CG); ↑ Travel speed (EG > CG);
Xiquan (2021) [4]	*n* = 28 Chronic stroke patients	EG: weight bearing and standing balance training, 10~15 RM each time; 3 times/week, 4 weeks in total; CG: Physical therapy; 50 min/time, 5 times/week, 4 weeks in total	Muscle strength, maximum weight bearing, MAS, walking speed, PGIC	↑ EG: maximum weight bearing > CG ↑ EG: Muscle strength lower limb ↑ EG: Walking speed >CG ↑ EG: PGIC > CG
Li, et al., (2021) [33]	*n* = 20 Chronic stroke patients	EG: 15 min warm up, 30 min 10 RM, 45 min/time, 3 times/week, 6 weeks in total; CG: Passive ROM traction, 3 times/week, 6 weeks.	Muscle trength; Walking and stair walking speed; SF-36	↑ Strength in both (PRE > CG); ↑ Walking speed in both group; SF-36 (EG ↔ CG).

Note Abbreviations: 6 MW: 6-min walk; 2 MW: 2-min walk; AROM: Active range of motion; BBS: Berg Balance Scale; GDS: Geriatric depression scale; LLFDI: Late-life function and disability instrument; MAS: Motor assessment scale; PGIC: Patient global impression of change; RM: Repetition maximum; SIP: Sickness impact profile; SIS: Stroke impact scale; TUG: Timed Up & Go; EG: Experience Group; CG: Control Group; SF-36: Health Related Quality of Life, Short Form-36. WMFT: Wolf Motor Function Test; TRT: Task-related Training; RMA: Rivermead Stroke Assessment; FT: Functional Task Practice; ↔: no difference between the two groups; FMA: Fugl-Meyer Assessment; ↑: increase; ↓:decrease.

**Table 2 nutrients-14-04995-t002:** Literature on the benefits of protein or other nutritional supplement interventions for stroke patients/elderly population.

Author/Year	Test Group (Number)	Sports Training Program	Nutrition/Protein Supplement	Key Findings
Rabadi et al., (2008) [13]	*n* = 116 Patients with acute stroke	Hospitalized rehabilitation treatment for acute strok	Nutrition supplement: 3 times/day, 5 g protein/time.	↑ 2 min and 6 min walking performance; ↑ Independent functional pointer (FIM); ↑ FIM motor subscore
Scherbakov et al., (2016) [56]	*n* = 110 Patients with acute stroke	Rehabilitation treatment for acute stroke patients during hospitalization (<8weeks after onset)	Essential amino acid supplement (4 g essential amino acid); 3 times/day for 8 weeks	Nutritional supplement strategy could significantly improve the muscle atrophy of CVA patients.
Rundqvist et al., (2017) [57]	*n* = 31 Patients with acute stroke	Curling rehabilitation therapy exercise for patients with acute stroke	Nasogastric tube enteral nutrition formula containing casein; Nasogastric tube enteral nutrition formula containing whey protein	Nasogastric tube enteral nutrition formula containing whey protein can more effectively reduce systemic inflammatory reaction and increase antioxidant capacity in elderly patients with acute stroke
Takeuchi et al., (2019) [58]	*n* = 42 Patients with acute stroke	Progressive resistance movement, balance movement; 60 min/time, 3 times/week, 9 months in total.	Supplemented twice a day (at 10:00 and 16:00 respectively): 15 g protein, 25 g carbohydrate, 4.4 g fat (200 kcal in total).	After 3 months, muscle strength ↑ 57%; Simple exercise intervention did not cause difference in lower limb muscle strength; Protein supplement and exercise intervention did not cause FFM difference
Carlsson et al., (2011) [59]	*n* = 177 Patients with acute stroke	High intensity exercise for 3 months; Urgent supplement after exercise	EG: protein supplement 7.4 g, carbohydrate 15.7 g, fat 0.43 g; CG: placebo supplement, containing 0.2 g protein and 10.8 g carbohydrate	There was no difference in muscle mass and body weight; The muscle mass and body weight of EG were negatively affected after 6 months; Inadequate nutrition supplement compensates the energy required for high-intensity exercise.
Jiayue, et al., (2017) [5]	*n* = 12 Patients with acute stroke	Divide into two groups of six; 12 week resistance movement	Group 1: 0.8 g protein/person/day/1KG body weight; The second group: 1.6 g/person/day/1KG body weight.	Resistance exercise significantly increased the energy intake required to maintain weight by 15%, and reduced body fat by 2.2%; Protein intake does not make a difference
Esmarck et al., (2001) [60]	*n* = 13 Patients with acute stroke	Progressive resistance movement; 3 times/week, 12 weeks in total	Nutrition supplement: 10g protein, 7g carbohydrate and 3.3g lipid of milk and beans; In group P1, the intake time was immediately after exercise; P2 group: ingested 2 h after exercise	Cross sectional area of quadriceps femoris in P_1_ after exercise increased by 6.77%, and the average muscle fiber range increased by 24.01%
Zielinska, et al., (2021) [15]	*n* = 100 Patients with acute stroke	High intensity progressive resistance exercise, 3 times/week, 10 weeks in total	240 mL of soybean protein was supplemented every night, including 60% carbohydrate, 23% fat and 17% protein.	Exercise group: lower limb muscle strength improved by 113%; Walking speed increased by 11.8%; Step climbing force improved by 28.4%; CG: walking speed decreased by 1%; Nutritional supplements did not result in any significant differences in major outcomes.

## Data Availability

The original contributions presented in the study are included in the article, further inquiries can be directed to the corresponding authors.

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
