# Peer review of "A Review of Rehabilitation Benefits of Exercise Training Combined with Nutrition Supplement for Improving Protein Synthesis and Skeletal Muscle Strength in Patients with Cerebral Stroke"

_nutrients, 2022, doi:10.3390/nu14234995_

Round 1
Reviewer 1 Report
The strength of the paper is that it describes very specific medical care area for chronic stroke patients: rheabilitation through improved nutrition and exercise training. The another strength includes relatively broad and intensive database search that include six sources of publications covering more than 20 years. More strength has been added by attempting to explain molecular mechanisms of positive effects of exercise training in concert with nutritional supplements with plant proteins in promoting muscle growth/recovery. However, several long parts of the text are required to meet any proper citations due to context of discussion (lines 384-398, and 401-407, 426-450). Despite extended discussion,a couple of key problems directly related to the main scope of the review paper, have been omitted and not discussed. They include nutritional factors for patients with hip bone density/osteoporosis problems (as discussed by Li S.K.Y. et L., 2017, not cited in the paper) and nutritional status evaluations (for example, Serra M.C., 2018) that may also impact recovery of chronic stroke patients.
Author Response
Reply:
1)Thank the experts for their high affirmation of the quality of this study. Experts mentioned that several references in the article need to be supplemented. This revised version has been supplemented in appropriate places.
2) From 384-398 to 426-450, this is the author's summary and summary of previous literature research, and naturally there is no reference cited. For 426-450, relevant references are supplemented;
3)With regard to the nutritional factors and nutritional status assessment of patients with osteoporosis, the author added Li, SYY et al (2017) and Serra MC; Et al (2018) literature.
Reviewer 2 Report
The review manuscript by Liu et al. aimed to summarize the current progress in combining exercise rehabilitation and nutrition supplement intervention to improve the health of cerebral vascular accident (CVA) patients. Such a systematic review in this area will bring important insights and guidance for clinical workers and care givers.
Here lists a few of my concerns that could improve the overall quality of the manuscript if being addressed appropriately.
Q1: The overall brevity and conciseness of the context needs to be dramatically improved.
A comprehensive review of the topic does not necessarily mean that it will list all the details of every aspect of the referred peer studies, and certainly not relying on extending more pages based on long sentences and paragraphs. I believe if the authors can shorten the manuscript and focus on the main points, it will improve the readability for the audience.
Q1.1 Please consider switching long sentences into short sentences to improve the readability across the whole manuscript, and an example is Line 230-232.
Q1.2 Please consider shortening the long paragraphs by only including key relevant studies, and an example is Line 51-82
Q2: Please consider removing category sign like 1) from the section of “3. Results”. Such category sign may be good in the section of “2. Methods”, but will reduce the readability of the whole manuscript. Examples are found in sections of 3.1.3; 3.2.1; 4; 5.
Q3: Please consider removing the cartoon and human figures to improve the professionalism of scientific presentation in Figure 2. The authors should make sure they have the copyright to use these figures, especially a consensus form for using real human images in the publication. Otherwise, a self-drawn cartoon figures or simply using words are better options to present the same message.
Q4: I would suggest keeping the section of “Conclusion” in the end and before the section of “Problems and Prospects”. It’s better to leave the readers with key takeaways. For the final “Conclusion” section, please remove the first paragraph, since it is not a direct conclusion from this manuscript.
Q5: Line 124: What is PEDro Scale? A short explanation should be briefly described.
Author Response
Major points:
- It would be helpful if the authors provide a table summarizing the results from searching respective databases including CNKI and Weibo, e.g. how many papers are overlapping, the difference between results obtained from the Chinese search and results obtained from the English search, the shortlist and exclusion rate for each database.
Reply: Because Figure 1 gives a clear account of the literature shortlisting, adding a literature shortlisting and exclusion table is of little value, but it increases the length of the article. I hope the experts will have an insight.
- Fig.2 is not good enough. The authors should provide a figure legend and refine the figure. e.g. assign different colors to different components, avoid overlapping of arrows, give credits to the source of cartoons/photos, standardize the labeling and font size.
Reply: According to the suggestions provided by the experts, this revision has appropriately modified the drawing. This drawing is designed by the author himself. As for the cartoon/photo in the drawing, there is no patent dispute and signature problem. Thank you for your reminding.
- As this review focus on patients with cerebral stroke, in the part of prospects, the authors should consider discussing more on how nutrient supplements or exercise alone, as well as a combinational therapy of nutrient supplements and exercise can impact the neuron growth and quality and thus alleviate the harmful effects of cerebral stroke. e.g. through stimulating which intracellular process and which signaling pathway, what kind of pre-clinical models could be employed for the study in this scenario?
Reply: As a small review, it is impossible to write all the questions clearly, and the literature that can be collected is also limited. The author can only comment on the research that will be learned within the current scope of competence. I hope the experts will have an insight.
Minor points:
- Abstract lines 28-29, "This paper suggests that the future research direction should be the supplement of natural antioxidants, probiotics, compound nutritional supplements, and integrated human clinical research." is not very clear.
Reply: The expert opinion is very pertinent and has been revised.
- The authors should consider using more standard terms throughout the manuscript, e.g. line 209, use "pathway" instead of "message path".
To summarize, the topic of the review is important and the paper may be relevant for Nutrients. The manuscript covers key discoveries in this field and could be improved by addressing the above issues, I, therefore recommend reconsideration of the manuscript after major revision.
Reply: Thanks for the affirmation and encouragement of experts, and the standardization of relevant terms has been checked and modified.

Reviewer 3 Report
In this manuscript, the authors reviewed key literature regarding the benefits of exercise and nutrition supplements on the rehabilitation of patients with cerebral stroke, with an emphasis on the impacts of exercise and nutrition supplements in promoting protein synthesis and strength of skeletal muscles. The useful compilation uncovered in this manuscript thus provides inspiring insights for both the scientific community and the practitioners.
However, the author should consider addressing the following issues in this manuscript:
Major points:
1. It would be helpful if the authors provide a table summarizing the results from searching respective databases including CNKI and Weibo, e.g. how many papers are overlapping, the difference between results obtained from the Chinese search and results obtained from the English search, the shortlist and exclusion rate for each database.
2. Fig.2 is not good enough. The authors should provide a figure legend and refine the figure. e.g. assign different colors to different components, avoid overlapping of arrows, give credits to the source of cartoons/photos, standardize the labeling and font size.
3. As this review focus on patients with cerebral stroke, in the part of prospects, the authors should consider discussing more on how nutrient supplements or exercise alone, as well as a combinational therapy of nutrient supplements and exercise can impact the neuron growth and quality and thus alleviate the harmful effects of cerebral stroke. e.g. through stimulating which intracellular process and which signaling pathway, what kind of pre-clinical models could be employed for the study in this scenario?
Minor points:
1. Abstract lines 28-29, "This paper suggests that the future research direction should be the supplement of natural antioxidants, probiotics, compound nutritional supplements, and integrated human clinical research." is not very clear.
2. The authors should consider using more standard terms throughout the manuscript, e.g. line 209, use "pathway" instead of "message path".
To summarize, the topic of the review is important and the paper may be relevant for Nutrients. The manuscript covers key discoveries in this field and could be improved by addressing the above issues, I, therefore recommend reconsideration of the manuscript after major revision.
Author Response

(The authors gave the same response as above.)

Reviewer 4 Report
Liu and colleagues reviewed recent literatures that may have relevance to muscular rehabilitation treatment of stroke patients with the emphasis on supplementing exercise with nutritional and other health-promoting additives as well as probiotics. Overall, the review may be of interest to those involved in rehabilitation of stroke patients. However, many of the statements sound similar and in many places, descriptions are vague. For instance, the authrs describe the protein, amino acid, carbohydrate, vitamin, and probiotic supplements. What is critical are the molecular identity (by specific name) of proteins, amino acids and all other additives and how they were used (how much, how often, how long, and all other available information in reviewed papers). In addition, how many patients/volunteers were involved, and what kind volunteers? If the data come from animal experiments, that should be indicated. It seems to make sense if a table is provided for each section containing all the information from the papers reviewed. Such a table should contain all the specific information from each study. In addition, some data come from the field of sports medicine. Subjects in such studies are in general much younger and physically better-fit than most of the stroke patients. Thus, it is possible that what works in sports medicine may not be applicable to stroke patients. There are some issues regarding literature citation. First and foremost, please make sure that the correct papers are cited in the text. Second, there are 78 papers listed, but the last citation number in the text is 76.
Listed below are specific comments.
1. In Abstract (lines 18-19). The authors state, "... conducted a preliminary study on the effects of various nutrition supplement strategies on stroke rehabilitation." This statement indicates that the authors performed some experiments. If so data from such a study must be presented in this article.
2. Introduction (lines 61-62). What is the difference between "therapeutic exercise" and "rehabilitation exercise"? Please explain.
3. Line 100. "... oxidative stress, antioxidants, probiotics, balance, etc." These are keywords used to select appropriate papers. What does "etc." mean? Were there other keywords not listed here used? If so, they must be listed. In a strict scientific statement, one should not use "etc." This makes the statement umbiguous. In fact, the authors use "etc." in many places throughout the manuscript including in a figure. Where "etc." is used, please make the list or statement complete so as to avoid making readers wondering and frustrated.
4. Line 114. The literatures reviewed were published in either English or Chinese. In the reference list, there are no Chinese papers. How were the Chinese papers dealt with? How can readers see the origical papers in Chinese? What information and data were taken from Chinese papers? By the way, I cannot locate Ref. 2.
5. Lines 130-133. Papers were evaluated by PEDro scale and were grouped into the high-quality and the low-quality papers. Did the authors use only the high-quality papers? Or were both groups of papers used? If the latter, how was the quality of paper reflected making conclusions in the review?
6. Line 209. PI3K is phosphoinositide 3 kinase.
7. Line 216. "....increasing protin synthesis." What proteins? How would you know that these are critical for improving the skeletal muscle function? How was this meaured?
8. Line 225. What is long-term exercise? This is vague.
9. Line 269. What is long-term exercise here? Are the long-term exercises in 8 and 9 the same in terms of lengths of exercise (time/session), duration (how many days) and all other conditions?
10. Lines 283-288. These two phrases are not sentences.
11. Line 289. "The results" What results?
12. Lines 293-294. How lonng were these experiments done? The general methods are not clearly described.
13. Lines 299-302. These percentages mean little as there are no comparisons made with appropriate controls.
14. Lines 332-335. This is a good summary of the work.
15. Line 425. What does "muscle protein quality" mean?
16. References. Please check that the correct papers are cited throughout the text. How were papers written in Chinese handled in the review. If they are not available to the general readers, they should not be included in the review.
Author Response
Reply: Thank you very much for the experts' affirmation of the advantages of this article and the lack of research. This revision will strictly implement the experts' suggestions. The number of selected articles in this article is 68 (see Figure 1), and clerical errors may occur during translation (78). However, the number of references at the end of the article is not necessarily equal to the number of selected articles, and the latter is certainly more than the former.
Listed below are specific comments.
- In Abstract (lines 18-19). The authors state, "... conducted a preliminary study on the effects of various nutrition supplement strategies on stroke rehabilitation." This statement indicates that the authors performed some experiments. If so data from such a study must be presented in this article.
Reply: The expert construction is very pertinent. Here is an error in translation and expression, which has been corrected.
- Introduction (lines 61-62). What is the difference between "therapeutic exercise" and "rehabilitation exercise"? Please explain.
Reply: This is a mistake in the author's translation. There is no such two concepts, and their essence is consistent. Has been modified.
- Line 100. "... oxidative stress, antioxidants, probiotics, balance, etc." These are keywords used to select appropriate papers. What does "etc." mean? Were there other keywords not listed here used? If so, they must be listed. In a strict scientific statement, one should not use "etc." This makes the statement umbiguous. In fact, the authors use "etc." in many places throughout the manuscript including in a figure. Where "etc." is used, please make the list or statement complete so as to avoid making readers wondering and frustrated.
Reply: The expert's criticism is completely correct. The author of this revision has checked the full text against the problem raised by the expert and made corresponding amendments.
- Line 114. The literatures reviewed were published in either English or Chinese. In the reference list, there are no Chinese papers. How were the Chinese papers dealt with? How can readers see the origical papers in Chinese? What information and data were taken from Chinese papers? By the way, I cannot locate Ref. 2.
Reply: The experts' criticism is very pertinent. Only 6 Chinese literatures were shortlisted in this review, including 2 references.
- Lines 130-133. Papers were evaluated by PEDro scale and were grouped into the high-quality and the low-quality papers. Did the authors use only the high-quality papers? Or were both groups of papers used? If the latter, how was the quality of paper reflected making conclusions in the review?
Reply: The score of the papers shortlisted in this study is ≥ 4.
- Line 209. PI3K is phosphoinositide 3 kinase
Reply: PI3K is phosphoinositide 3 kinase There is nothing wrong with this
- Line 216. "....increasing protin synthesis." What proteins? How would you know that these are critical for improving the skeletal muscle function? How was this meaured?
Reply: The increase of protein synthesis here, of course, refers to muscle protein. As for how to measure it, the method is already mature, so it will not be explained here.
- Line 225. What is long-term exercise? This is vague.
Reply: Regarding the duration of exercise intervention, this paper defines four weeks as short-term, four to eight weeks as medium-term, eight to 12 weeks as medium to long-term, and 12 weeks as long-term exercise.
- Line 269. What is long-term exercise here? Are the long-term exercises in 8 and 9 the same in terms of lengths of exercise (time/session), duration (how many days) and all other conditions?
Reply: In the exercise intervention prescriptions shortlisted in this study, there are great differences in the time and treatment of each exercise because of the great differences in the physical fitness of different patients. However, intervention prescription treatment is generally divided into short, medium, medium and long term. Each exercise lasted more than 30 minutes on average, and was divided into 4-8 groups, with an interval of 2-5 minutes between groups.
- Lines 283-288. These two phrases are not sentences.
第283-288行这两个短语不是句子
Reply: Appropriate modifications have been made
- Line 289. "The results" What results?
第289行“结果”结果是什么?
Reply: Appropriate modifications have been made
- Lines 293-294. How long were these experiments done? The general methods are not clearly described.
Reply: Relevant intervention methods have been supplemented
- Lines 299-302. These percentages mean little as there are no comparisons made with appropriate controls.
Reply: No comparison is required, but these data can provide reference for subsequent scholars to conduct similar experiments.
- Lines 332-335. This is a good summary of the work.
Reply: Thank you for your praise
- Line 425. What does "muscle protein quality" mean?
Reply: This is a translation slip. It should be the “muscle protein content”
- References. Please check that the correct papers are cited throughout the text. How were papers written in Chinese handled in the review. If they are not available to the general readers, they should not be included in the review.
Reply: Thanks for the expert's reminding. The author has reviewed the full text references, deleted and supplemented them as required.

Round 2
Reviewer 3 Report
In this version, the authors have successfully addressed both minor issues, and partially major issue 2.
However, the author failed to address the other two major issues mentioned in the previous comments, including:
Major points:
1: incorporating a table summarizing the results from searching respective databases including CNKI and Weibo.
3. adding in discussion on the potential beneficial impact of nutrient supplements and exercise on neuron growth and quality.
Further comments:
Major point 1: This point is similar to the comment 4 and 5 raised by Reviewer 4 and should be regarded as relevant to the quality of the study. The authors could also consider providing the relevant information in a supplementary table, so that the length of the main text would not be affected.
Major point 3: The authors should consider modifying the review title to a more specific one, “A Review Of Rehabilitation Benefits Of Exercise Training Combined With Nutrition Supplement For Improving Protein Synthesis and Skeletal Muscle Strength in Patients With Cerebral Stroke.”
To summarize, the authors have made several reasonable modifications to the previous version and the manuscript could be improved by addressing the remaining issues stated above, I therefore recommend acceptance of the manuscript after major revision.
Author Response
1: incorporating a table summarizing the results from searching respective databases including CNKI and Weibo.
Reply: As required, we have added the information table of important literature related to literature shortlisting.
- adding in discussion on the potential beneficial impact of nutrient supplements and exercise on neuron growth and quality.
Reply: Discussion on the potential benefits of nutritional supplements and exercise on the growth and quality of neurons
Further comments:
Major point 1: This point is similar to the comment 4 and 5 raised by Reviewer 4 and should be regarded as relevant to the quality of the study. The authors could also consider providing the relevant information in a supplementary table, so that the length of the main text would not be affected.
Reply: Relevant important contents have been improved according to experts' opinions.
Major point 3: The authors should consider modifying the review title to a more specific one, “A Review Of Rehabilitation Benefits Of Exercise Training Combined With Nutrition Supplement For Improving Protein Synthesis and Skeletal Muscle Strength in Patients With Cerebral Stroke.”
Reply: It has been modified as required by experts.
